# Household food insecurity, living conditions, and individual sense of security: A cross-sectional survey among Burkina Faso refugees in Ghana

Abdul-Wahab Inusah[1], Ken Brackstone[2], Tahiru Issahaku Ahmed[3], David Tetteh Nartey[1], Jessica L. Boxall[2], Ashley I. Heinson[2], Michael Head[2,4,5‡]*, Shamsu-Deen Ziblim[6‡]

1 JSI Research & Training Institute, INC., Takoradi, Ghana, 2 Clinical Informatics Research Unit, Faculty of Medicine, University of Southampton, Southampton, United Kingdom, 3 Bawku West District Assembly, Bolgatanga, Upper-East Region, Ghana, 4 School of Medicine, University for Development Studies, Northern Region, Ghana, 5 F.N. Binka School of Public Health, University of Health and Allied Sciences, Hohoe, Volta Region, Ghana, 6 Directorate of Academic Planning & Quality Assurance (DAPQA) University for Development Studies, Tamale, Ghana

‡ MH and SDZ are joint senior authors on this work.
* m.head@soton.ac.uk

## Abstract

Food insecurity and achieving adequate nutrition is a major global challenge, especially in vulnerable groups such as refugee communities. In West Africa, thousands of Burkina Faso refugees have crossed the border into northern Ghana due to conflict and instability in their home country. We conducted a one-off cross-sectional survey to assess household food insecurity, living conditions, and sense of security among Burkina Faso refugees currently residing in the Upper East region of Ghana. Study data was collected over 14–21 October 2022 from 498 refugee households, via registered refugee households who were contacted with the help of the community refugee focal persons. We used the validated USAID household food insecurity questionnaire, analysed using Rasch modelling, with descriptive statistics, and linear regression analyses (with significance at $p < 0.05$) to assess food insecurity. Results revealed that 100% of households experienced food insecurity, with 95.2% defined as moderate or severe, and 70.4% as experiencing severe food insecurity. Refugees from rural areas were less insecure compared to urban refugees (β = -4.25 [CI: -5.79 —-2.71], $p < .001$). Refugees residing in host communities experienced lower food insecurity than those in designated refugee camps (β = -1.56 [CI: -2.74 —-0.39,] $p = .009$). Further, refugees who were dissatisfied with their accommodation size were more likely to experience food insecurity (β = 2.96 [CI: -0.06–2.47], $p = .060$). Most refugees (73.5%) felt safe and welcomed by host communities. Our results highlight the extremely high prevalence of moderate and severe food insecurity in this vulnerable refugee population. We provide evidence to support the need to improve food distribution logistics, prioritising suitable accommodation, and facilitating access to healthcare. Follow-up research, such as repeated community

**Data Availability Statement:** The dataset is publicly available on the Figshare repository at https://doi.org/10.6084/m9.figshare.23782893.

**Funding:** The Clinical Informatics Research Unit (https://www.the-ciru.com/), University of Southampton, provided a business development award of £750 to author MGH. The funder played no role in the study design, data collection and analysis, decision to publish, or preparation of the manuscript.

**Competing interests:** The authors have declared that no competing interests exist.

surveys, can track this evolving situation to continuously inform decision-making for refugee support.

## Introduction

Food insecurity leads to malnutrition, which negatively impacts upon all individuals, but can have a devastating effect on more vulnerable groups such as young children and pregnant women [1, 2]. It is estimated that 1 in 3 people (2.3 billion) experienced some level of food insecurity in 2021 [3], with increases in the prevalence of undernourishment observed in sub-Saharan Africa [4]. The Food and Agriculture Organisation reported that the prevalence of moderate or severe food insecurity across Africa was 58.9% across 2020–2022 [3]. Similarly, the Global Nutrition Report highlights that no African country is on track to meet any diet-related non-communicable disease targets [5]. This makes it challenging, if not impossible, to achieve the Sustainable Development Goals target to eradicate hunger by 2030 [6].

Globally, approximately 26 million people are currently refugees, with close to 80% of this population experiencing food insecurity regardless of location [1]. Using food and nutrition security theory, there are four pillars of food insecurity that can help to understand why the disruption to a safe and balanced diet has come about: availability of food, accessibility, affordability, and the ability of a household to utilize the resources they have acquired to provide sufficient nutrition [7]. Violence can directly lead to food insecurity and malnutrition by obliterating food systems, thus affecting availability and accessibility; reducing farming labour can hinder availability and affordability; and destroying transport systems can prevent accessibility, which can lead to a reduction in community resilience [8–10].

Refugees often flee from armed conflict, leaving behind their homes and livelihoods. This presents difficulties by way of affording food, and many are no longer able to utilize food properly without a safe and equipped environment to prepare meals. Fleeing to another nation may also affect what foods can be grown or acquired. The inability to consume traditional and preferred foods is another issue of food insecurity that should not be forgotten in the interests of cultural preservation.

Burkina Faso is in West Africa, with an estimated population of 22.1 million. It is a landlocked nation bordering several other countries, including Ghana to the south. Since 2015, Burkina Faso has witnessed several waves of armed conflicts emanating from non-state armed groups, whose aims are to instrumentalize communal tensions and exploit the country's vulnerability. This has resulted in a displacement surge among its citizens [1]. Reports from the United Nations Childrens Fund (UNICEF) indicate that Burkina Faso recorded 488 armed attacks with 144 people killed in the first quarter of 2022 [11]. As a result of this conflict, the number of internally displaced persons (IDP) has risen to over 1.8 million, with 61% of the group being children [12]. The ongoing instability and violence means that there are declining crop yields, with a continued rise in prices of goods and services [11]. As a result, around 13 million people are faced with food insecurity, and one-third are malnourished children aged under 5 years old [13].

The insecurity situation in the country has resulted in the closure of 179 healthcare centres in the hardest hit regions, with 353 healthcare centres operating at suboptimal levels. These closures have denied over 2 million citizens access to health care [11]. A significant proportion of the population now have little access to basic social amenities such as water, food, shelter, education, and healthcare, with many people considering that their best option is to flee to

neighboring countries. As of February 2023, Ghana hosted 13,500 refugees, of which 4000 were Burkina Faso citizens seeking asylum [14]. These numbers continued to increase across 2022 and 2023, particularly within the Upper East Region of Ghana, which directly borders Burkina Faso [11]. The care of these refugees is the sole responsibility of the Ghanaian local and national authorities, and is supported by international humanitarian agencies such as United Nations High Commissioner for Refugees (UNHCR). Their arrival has not only resulted in increased pressure on Ghanaian and international agencies to provide humanitarian needs for these refugees, but also places excess pressure on social amenities in their host communities.

Living arrangements can significantly influence food security among refugees due to differences in access to resources, social support, and integration into host communities [1]. Refugees residing in local households may benefit from closer connections to local food systems, such as markets and community resources, and may gain access to share meals or household food supplies [15]. In contrast, refugees living in camp settings often face structural barriers, such a limited food availability, ration-based distributions, and dependency on humanitarian aid [16]. Thus, it is possible that there may be differences between those displaced populations who are residing in households inhabited by local residents compared with those in refugee camps or similar settings. Understanding these dynamics is crucial for designing targeted interventions that address food insecurity more effectively, while also guiding policy decisions on resource allocation for refugee support.

This study was conducted to assess the health, social, and economic needs of these refugees in the Upper East Region of Ghana, with a specific focus on food insecurity and the living conditions among this population. We aimed to explore and understand predictors of food insecurity, and to describe some of the key characteristics of this vulnerable group. Our hypothesis was that refugees residing in households inhabited by local residents would have higher levels of food security than those residing in refugee camp settings.

## Methods

### Study design and settings

A community-based cross-sectional study was implemented in Ghana using in-person data collection from household heads of identified refugees (S1 File). Data was collected in the rural districts of Binduri District, Bawku Municipal, and Bawku West, situated within the Upper East Region of Ghana (Fig 1), where the majority of displaced people are located. The region contains 15 administrative districts with a population of 1,301,226 [17], and a total landmass of 8,842sq/km constituting 2.7% of the country's land size [18]. The regional capital is Bolgatanga, approximately 60-100km west of the study sites. Upper East Region shares international borders in the north with Burkina Faso, and in the east with Togo [18]. The location of the region exposes it to cross-border activities with neighbouring migrants and traders from Burkina Faso and Mali, Togo and Niger using it as transmit point into Ghana.

### Study population and sample size determination

All consenting adult heads of refugee households in the host communities were eligible participants. Start and end dates of data collection was 14–21 October 2022. Sample size was calculated using a single population proportion formula with an assumption of 50% population proportion due to the unknown population size. With a 95% confidence level, a 5% margin of error, and a normal population distribution Z = 1.96, we adjusted for a 5% non-response rate. This provided an estimated sample size of 400. The research team was experienced at data collection with vulnerable populations [19, 20], and utilised the support and assistance of local authorities and local residents throughout the study to ensure sensitivity around our approaches.

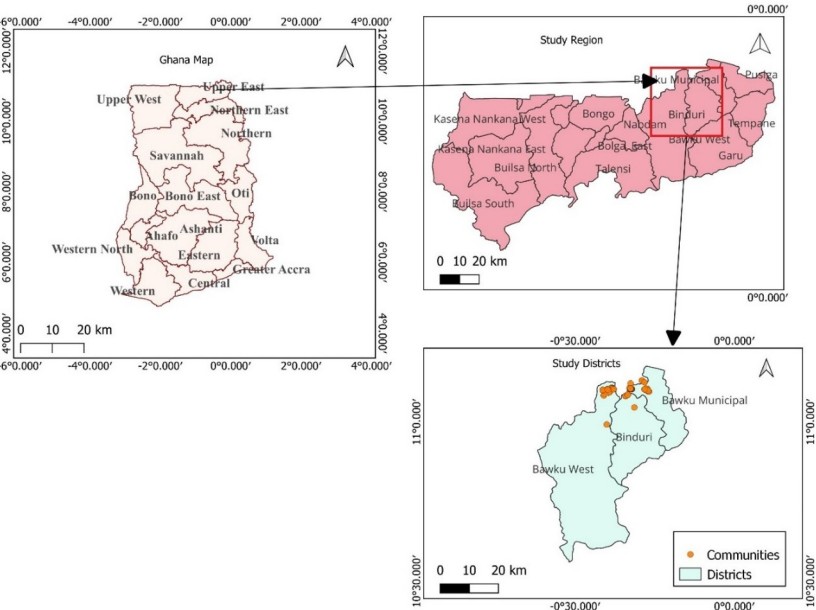

**Fig 1. Map showing the regions of Ghana, Upper-East Region, and the municipals with refugee communities.**
Image created in QGIS Version 3.28.1 by author Abdul-Wahab Inusah. Base map shapefile was downloaded from
DIVA-GIS via https://www.diva-gis.org/datadown.

## Sampling technique and sampling procedures

Using the list provided by the National Disaster Management Organization department of all
three administrative districts, the number of participants studied in each district was propor-
tional to the total number of registered refugees in the district. The number of registered refu-
gees for each district was 1072 in Binduri District (64%), 495 in Bawku Municipal (29%) and
112 in Bawku West district (7%), which totalled 1679 individuals. Since the calculated sample
size for the study was 400, the minimum preferred number of participants for each district was
256 for Binduri district, 116 for Bawku Municipal, and 28 for Bawku West district. A purpo-
sive sampling technique was used to select only communities hosting Burkina Faso refugees.
With the help of the community refugee focal persons (persons appointed by the respective
local assemblie) who documented all refugees in the communities, we identified and contacted
all registered refugee households using the refugee register for each community. In each
household, the household head was our respondent. A household was revisited in the absence
of the household head. If that individual was still not available upon the second visit, the survey
was carried out with an individual nominated by other household members.

Inclusion criteria was for adults aged 18 years and over from Burkina Faso, classified as ref-
ugees, and with the capacity and standard of health to provide informed consent. Exclusion
criteria covered those under 18 years, not refugees from Burkina Faso, and lacking capacity or
in very poor health.

## Study variables

**Household food insecurity assessment.**　We used the validated USAID household food
insecurity assessment score (HFIAS) module [21], which was used to assess household food
insecurity over the previous 30 days. HFIAS consist of 9 questions grouped into 3 domains,
including: (1) anxiety and uncertainty, (2) insufficient quality, and (3) insufficient food intake

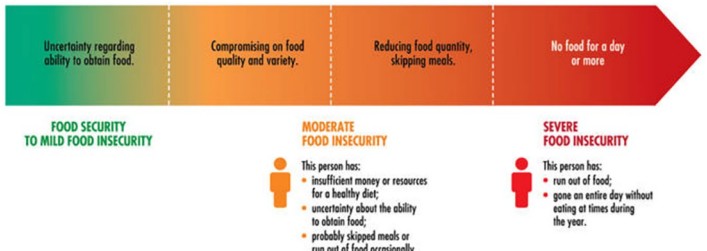

**Fig 2. Visual depiction of the different thresholds in the food insecurity experience scale.**

and physical consequences. Participants responded "yes" or "no" to each of the nine questions. A higher HFIAS score indicated greater food household insecurity (Fig 2) [21].

**Living condition variables.** Participants were asked about their current accommodation and living conditions, including whether they knew locals prior to arriving in the community, the size of their accommodation facility, and presence of any social support for their household and sources of these social support systems.

**Sense of security.** To measure the safety and sense of security among the study participants, 4 questions were asked using the 5-point Likert scale (e.g., "Generally, I feel satisfied with my current living conditions"; 1 = strongly disagree; 5 = strongly agree). Alongside satisfaction with living conditions, questions also covered perceptions of safety, and any indication of being welcomed by, or receiving hostility from, the local community. We averaged the items to form a single index ($M = 3.45$, $SD = 0.99$; $\alpha = 0.68$).

**Sociodemographic variables.** These included sex (male, female), age (continuous), marital status (not married, married, separated/divorced, widowed), religious affiliation of the household head (Christian, Muslim, Traditional, other), household size (number of people sharing a common shelter), number of children under five in the household, and number of pregnant women in participating households.

**Data collection procedure and quality control.** The questions were adapted from an existing questionnaire used in 2022 with Ukrainian refugees and internally-displaced persons by the same research team [22, 23]. These surveys were based upon existing literature, including the European health needs survey. The survey questions were uploaded to an Android smartphone App (ODK). The tool was pretested in Nabdam, a neighbouring district among 20 refugees. These test responses were not included in the final analysis. All the necessary adjustments, including skip logic, and age groupings were made before the main data collection began.

Ten community health workers participated in a two-day data collection training workshop, which covered issues such as data collection, good research practice, and the need for explaining the study and obtaining informed consent. As we anticipated low literacy rates, survey questions were verbally read out and the data collectors recorded their responses directly onto the ODK app. The survey was written in English. A local dialect, Moshie, was used for verbal translations and context.

## Data analysis techniques

Rasch modelling was undertaken in R 4.3.2 using the package "RM.Weights", as the recommended analysis protocol by the FAO [24]. The dataset was pre-processed to ensure compatibility with Rasch modelling and a subset that contained only the questions from the questionnaire that related to the variables of FIES 1–8 were included for a non-weighted

method of Rasch modelling [25]. The model determined the prevalence of moderate or severe food insecurity, and the prevalence of severe food insecurity as percentages. Following the generation of food deprivation severity score data visualisation was undertaken using summary statistics to represent this population's food severity in a visual, easily interpretable manner. The coding from R 4.3.2 is available (S2 File).

Stata version 15 was utilized in carrying out descriptive and inferential statistics for the data. Descriptive statistics included frequencies, percentages, mean and standard deviation. A reliability test was conducted for the HFIAS ($\alpha = 0.84$) and sense of security ($\alpha = 0.68$) items, which indicated acceptable to excellent reliability. We checked for model fitness using the R-squared ($R^2$) value to determine how much variance of our outcome variable was explained by the independent variables. The observed $R^2$ value for this study was 0.83 indicating a very good model fitness. We calculated the variance inflation factor (VIF) for each of the predicting variables and all were below 0.3 indicating no multicollinearity. To assess determinants of household food insecurity (our key outcome variable), a linear regression analysis was conducted, and p-values of less than 0.05 were considered statistically significant. We first performed a univariate linear regression analysis and only candidate variables with p-values less than 0.05 were included in the final multivariable linear regression model.

### Ethical consideration

Ethical approval (UDS/RB/112/22) was obtained from the University for Development Studies Research and Ethics Review (S3 File). Separately, approval and community entry was sought from the three district assemblies to carry out the study within their populations. Informed written consent was obtained from all individuals before they participated in the study. Potential participants were advised they did not have to take part in the study, and that they would not face any penalty or prejudice by their withdrawal from the study or unwillingness to participate.

## Results

In total, 498 participants completed the survey (Table 1), with a majority being female (65.3%). The average age was 37.2 years (*SD* = 16.2, *Range* = 18–96). Nearly 86.0% had no formal education, 93.0% were affiliated to Islamic religion, and 90.0% indicated that their home community was a rural area (Table 1). The mean household size was 6.0 people, with a mean of 1.4 children aged under 5 years per household. Finally, 5.5% of the female group indicated that they were pregnant.

### Prevalence of household food insecurity among the study households

Of the 498 households served, all 498 (100.0%) of households were defined as experiencing moderate to severe food insecurity (Fig 3). Across the previous 4 weeks, 35% of households were often worried that their household did not have enough food, and approximately 38% of households reported that they were unable to eat the kind of food the household preferred. Results also showed that 27.5% of households have family member(s) who have experienced a whole day and night without eating anything due food unavailability. The prevalence of moderate or severe food insecurity in this population was 95.2%, and the prevalence of severe was 70.4%. The median of the raw HFIAS score was 8 (lower quartile: 7, upper quartile: 8); thus, participants had answered yes to every question and experienced severe food insecurity. There was no significant difference in food insecurity between the geographical communities of Binduri District, Bawku Municipal, and Bawku West.

**Table 1. Sociodemographic characteristics of the Burkina Faso refugees in Ghana (*n* = 498).**

| Characteristics | Frequency | % |
|---|---|---|
| **Sex** | | |
| Female | 325 | 65.3 |
| Male | 173 | 34.7 |
| **Age [Years]** | | |
| 18–30 | 227 | 45.6 |
| 31–40 | 115 | 23.1 |
| 41–50 | 45 | 9.0 |
| 50–60 | 57 | 11.4 |
| 61+ | 54 | 10.8 |
| **Marital status** | | |
| Never married | 35 | 7.0 |
| Divorced/widowed | 70 | 14.1 |
| Married | 393 | 78.9 |
| **Religious affiliation** | | |
| Christian | 29 | 5.8 |
| Traditional | 7 | 1.4 |
| Islam | 462 | 92.8 |
| **Educational level** | | |
| No formal education | 426 | 85.5 |
| Basic secondary school | 63 | 12.7 |
| Complete secondary school | 9 | 1.8 |
| **Description of participants home community** | | |
| Countryside/rural | 448 | 90.0 |
| Urban/city | 50 | 10.0 |
| **When did you arrive at your current location?** | | |
| 1–7 days ago | 9 | 1.8 |
| 2 weeks-1month ago | 60 | 12.1 |
| More than 1month ago | 429 | 86.1 |

The Food and Agricultural Organization calculated food insecurity for West Africa by including the countries of Benin, Burkina Faso, Cabo Verde, Cote D'Ivoire, Gambia, Ghana, Guinea, Guinea-Bissau, Liberia, Mali, Mauritania, Niger, Nigeria, Senegal, Sierra Leone, and Togo [3]. The refugee moderate and severe food insecurity was higher than the West Africa average, and the Burkina Faso national figure (Table 2) [3].

## Determinants of household food insecurity

This study examined predictors of food insecurity among Burkina Faso refugees living in Ghana using univariate and multivariate linear regression models. In the univariate models, gender and age group were not significantly associated with food insecurity. However, location prior to migration and current housing situation were significant predictors. Specifically, refugees who lived in urban areas before migrating reported significantly higher food insecurity scores compared to those from rural areas (β = 4.56, 95% CI: 3.05–6.07, $p$ <0.001). Compared to living in housing specifically set up for refugees (A), living with somebody the refugee knew before, such as relatives or friends (C), was associated with lower food insecurity scores (β = -2.45, 95% CI: -3.60 –-1.31, $p$ <0.001). Finally, refugees who were happy with their living space

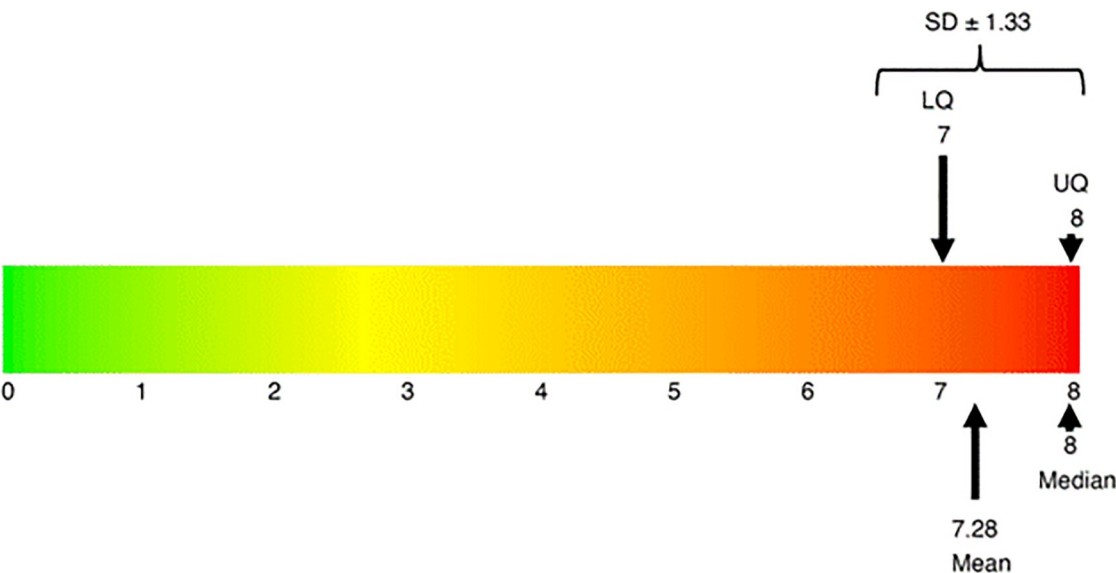

**Fig 3. Average score of food insecurity experience scale in this population.** This figure mimics the Food and Agricultural Organization scale of mild to severe food insecurity, demonstrating the mean and median scores fall within the moderate category. LQ is the lower quartile, and UQ is the upper quartile.

had lower food insecurity scores than those unhappy with their space (β = -2.89, 95% CI: -4.88 –-0.91, $p$ = 0.004).

The multivariate model accounted for potential confounding between predictors. Location prior to migration (β = 4.33, 95% CI: 2.78–5.88, $p$ <0.001) and living with somebody the refugee knew before (Camp C) (β = -1.62, 95% CI: -2.79 –-0.45, $p$ = 0.007) remained significantly associated with food insecurity. Happiness with living space also remained significant, with a slightly attenuated effect size (β = -2.16, 95% CI: -4.10–0.22, $p$ = 0.03). Results are shown in Table 3.

## Living conditions of the refugees

By accommodation status, 54.0% were living in housing specifically set up to house refugees, with 22.7% living with somebody they knew before migrating to Ghana, such as relatives or friends (Table 4). A further 19.5% were living in accommodation provided by a member of the host community who was not previously known to the refugee participant. Most females (69.1%) were living in housing specifically set up to house refugees. By living conditions,

**Table 2. Prevalence of food insecurity in this study population, compared to Burkina Faso's national prevalence and the global standard from the Gallup World Poll 2020–2022.**

|  | Prevalence of moderate or severe food insecurity (%) | Prevalence of severe food insecurity (%) |
|---|---|---|
| **Refugee study population** | 95.2 | 70.4 |
| **Burkina Faso (2020–22)** | 56.9 | 21.2 |
| **West Africa (2020–22)** | 64.1 | 21.2 |
| **Global standard (2020–22)** | 29.5 | 11.3 |

**Table 3. Multivariate linear regression model of predictors of food insecurity among Burkina Faso refugees in Ghana (n = 489).** $\beta$ = beta coefficient, t = test statistic.

| Variable | Univariate | | | Multivariate | | |
|---|---|---|---|---|---|---|
| | β (95% Cl) | t | p value | β (95% Cl) | t | p value |
| **Gender** | | | | | | |
| Female (ref.) | Ref. | | | Ref. | | |
| Male | -0.36 (-1.34–0.63) | -0.71 | 0.48 | -0.24 (-1.19–0.71) | -0.50 | 0.62 |
| **Age group** | | | | | | |
| 18-30yrs (ref.) | Ref. | | | Ref. | | |
| 31-40yrs | -0.54 (-1.73–0.66) | -0.88 | 0.38 | -0.45 (-1.59–0.69) | -0.78 | 0.44 |
| 41-50yrs | -0.92 (-2.40–0.57) | -1.22 | 0.23 | -0.74 (-2.16–0.68) | -1.03 | 0.31 |
| 50yrs+ | -0.07 (-1.36–1.23) | -0.10 | 0.92 | 0.57 (-0.68–1–82) | 0.90 | 0.37 |
| **Prior residence** | | | | | | |
| Countryside/rural (ref.) | Ref. | | | Ref. | | |
| Urban/city | 4.56(3.05–6.07) | 5.95 | <0.001 | 4.33 (2.78–5.88) | 5.50 | <0.001 |
| **Current residence** | | | | | | |
| A (ref.), in refugee housing | Ref. | | | Ref. | | |
| B, with local residents | 0.01 (-1.21–1.23) | 0.01 | 0.99 | 0.78 (-0.43–1.99) | 1.26 | 0.21 |
| C, with friends or relatives | -2.45 (-3.60 –-1.31) | -4.18 | <0.001 | -1.62 (-2.79 –-0.45) | -2.72 | 0.007 |
| D, with somebody who I was put in touch with | -1.43 (-387–1.01) | -1.15 | 0.25 | -0.59 (-2.97–1.79) | -0.49 | 0.63 |
| **Accommodation size** | | | | | | |
| Unhappy with space (ref.) | Ref. | | | | | |
| Happy with space | -2.89(-4.88 –-0.91) | -2.86 | 0.004 | -2.16(-4.10–0.22) | -2.19 | 0.03 |

A = In housing specifically set up to house refugees or internally displaced people(ref.), B = With a local person or people who have offered me accommodation,

C = With somebody I knew before, such as relatives or friends, D = With somebody with whom I was put in touch, for example, friends of friends

84.1% of participants were least happy about their current living conditions, with only 15.9% reporting being happy. In terms of support system, only 120 (24.1%) participants had received any welfare or housing payment to support their living standards (Table 2). Of the 120 participants who received financial support, most had received support from more than one source, with 94.2% receiving welfare payments from the Ghanaian Government, and 85.0% from international agencies such as UNICEF and UNHCR as their source of welfare payments (Fig

**Table 4. Living conditions of the Burkina Faso refugees in Ghana (n = 498).**

| Variable | Frequency | Percentage |
|---|---|---|
| **Where are you currently staying?** | | |
| In housing specifically set up to house refugees or internally displaced people | 269 | 54.0 |
| With a local person or people who have offered me accommodation | 97 | 19.5 |
| With somebody I knew before, such as relatives or friends | 113 | 22.7 |
| With somebody with whom I was put in touch, for example, friends of friends | 19 | 3.8 |
| **How would you describe the size of your accommodation?** | | |
| Happy | 79 | 15.9 |
| Unhappy | 419 | 84.1 |
| **Have you received any welfare or housing payments to support yourself since leaving Burkina Faso?** | | |
| No | 378 | 75.9 |
| Yes | 120 | 24.1 |

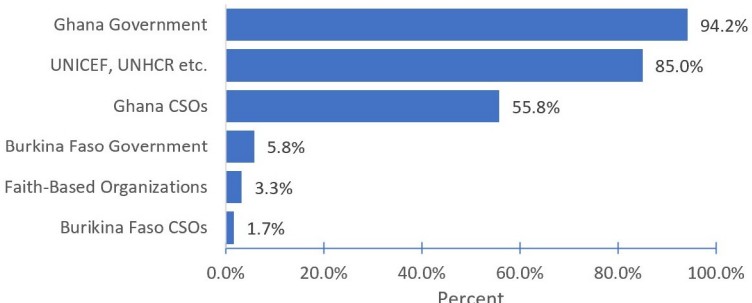

**Fig 4. Main sources of welfare or payment support among Burkina Faso asylum refugees in Ghana (n = 120, CSO = civic society organizations).**

4). Overall, 76.0% of respondents indicated they did not have enough financial support and other resources to meet their household basic needs.

## Self-reported sense of security in the current living conditions among Burkina Faso refugees

Regarding participants' sense of security, 71.9% of respondents agreed to some extent that they feel satisfied with their current living conditions. However, 28.5% disagreed to some extent that they feel satisfied with their current living conditions. When asked about community safety, 72.5% agreed to some extent that they felt safe in their new location, with 16.0% expressing some form of disagreement. By community response, 73.5% agreed to some extent that they felt welcome by the local community, whereas around 17.0% disagreed to some extent. Finally, participants 23.3% indicated that they had perceived some level of hostility or anger from others in their local community (Table 5).

## Discussion

Displaced persons are often disadvantaged with access to adequate food and suitable living conditions. Among Burkina Faso refugees currently living in the northern part of Ghana, our study found that 100% of our survey population experienced household food insecurity, with 95.2% of households experiencing moderate or severe insecurity. Self-reported satisfaction of living conditions was variable across the study population, with most participants reporting that they felt safe and welcomed by their new local community.

**Table 5. Sense of security in the current living conditions of the study participants, asked using a Likert scale where 1 for strongly disagree and 5 for strongly agree.**

| Variables | Responses (%) | | | | | Mean* |
|---|---|---|---|---|---|---|
| | Strongly disagree | Somewhat disagree | Uncertain | Somewhat agree | Strongly agree | |
| Q13 Generally, I feel satisfied with my current living conditions | 72 (24.5) | 20 (4.0) | 48 (9.6) | 129 (25.9) | 229 (46.0) | 3.8 |
| Q14 I feel safe in the place where I am currently | 54 (10.8) | 26 (5.2) | 57 (11.4) | 95 (19.1) | 266 (53.4) | 4.0 |
| Q15 I have been made to feel welcome by the local community | 49 (9.8) | 20 (4.0) | 63 (12.7) | 76 (15.3) | 290 (58.2) | 4.1 |
| Q16 My family and I have experienced hostility or anger from other people in the local community | 338 (67.9) | 22 (4.4) | 22 (4.4) | 63 (12.7) | 53 (10.6) | 1.9 |

* Mean refers to mean score across the 5-point Likert scale, where 1 for strongly disagree and 5 for strongly agree

The level of food insecurity was substantially higher than the national prevalence of food insecurity in Burkina Faso and across West Africa [26, 27]. However, there is likely to be significant sub-national variation. For example, a 2024 study found 61.5% prevalence of food insecurity from a rural community in the Northern Region of Ghana [28]. Among other refugee populations, food insecurity is typically very high. For example, several studies have reported greater than 80% insecurity for refugees in the USA, with 42% of children experiencing hunger [29]. Similar results were found among Syrian refugees in Canada [30], and Afghan refugees in Iran [31]. Lower insecurity was reported among Iraqi refugees in Lebanon (44.4%) and Scotland (56%) [32]. Our findings add to the evidence-base around risks of hunger and malnutrition among these vulnerable populations. There may be a need for targeted interventions to reduce the negative impact on the lives of under-5s, pregnant women, and older people, as these populations are disproportionately affected by food insecurity and conflict [30, 33, 34]. For those who remained in Burkina Faso, there is a longer-term trend of difficulties with food security as a result of climate change, the impact of COVID-19, increases in food prices adversely affecting purchasing power and affordability, and the continued rise in armed conflicts [33, 35, 36].

The conceptual framework in Fig 5 is adapted from adapted from the IPC Integrated Food Security and Nutrition Conceptual Framework [37]. It demonstrates the interactions between the pillars of food security, and the difficulties that conflict and displacement introduce. The findings from this study indicate that all pillars are affected; all variables deemed important in light of the results are highlighted. Participants make reference to food made available to them by their relatives, or via food aid and welfare packages in the short-term, as supporting their availability and accessibility of food. These are often the hardest pillars to address in the long-term deprived conditions due to a lack of storage facilities, income and housing instability, and intermittent aid [38]. Accessibility is also a prominent issue; not least because of the aforementioned issues, but also because large households and how food is allocated between members compromises the food security of those considered lowest priority in the hierarchy [39]. Living conditions and type of area the displaced persons now reside in were also significant predictors of food insecurity. This would have a huge effect on how food is able to be utilized, as refugees have to adjust from an urban-rural or dissimilar environment. Availability of traditional foods or preparation methods may be compromised; one of the findings from the questionnaire was that participants had been unable to access their preferred foods since being displaced.

A 2019 qualitative study reviewed Ghanaian and Liberian displaced populations within Ghana, concluding that social support, plus accessibility and accessibility are among the key factors needed to improve their security [40]. The participants in our study were newly-arrived into that community, and there is some evidence from Ghana that nutrition practice improves over time, potentially due to increased chances of access to health advice and social support [41]. To fulfill food security, all these variables must be addressed in a sustainable manner. This will, in the long-term, prevent further vulnerability to malnutrition and poor health outcomes, and thus promote resilience to future acute events.

Around half of the participants in this study resided in refugee camps set up by UNHCR and the local authorities, and these people reported less satisfaction and greater insecurity with the quality of their accommodation compared to participants who were staying with community members who they knew. This may highlight the increased vulnerability of forced migrants who have little knowledge of their new community, with issues such as familiarity and language being factors that may make it more difficult to access food, water, and healthcare [42].

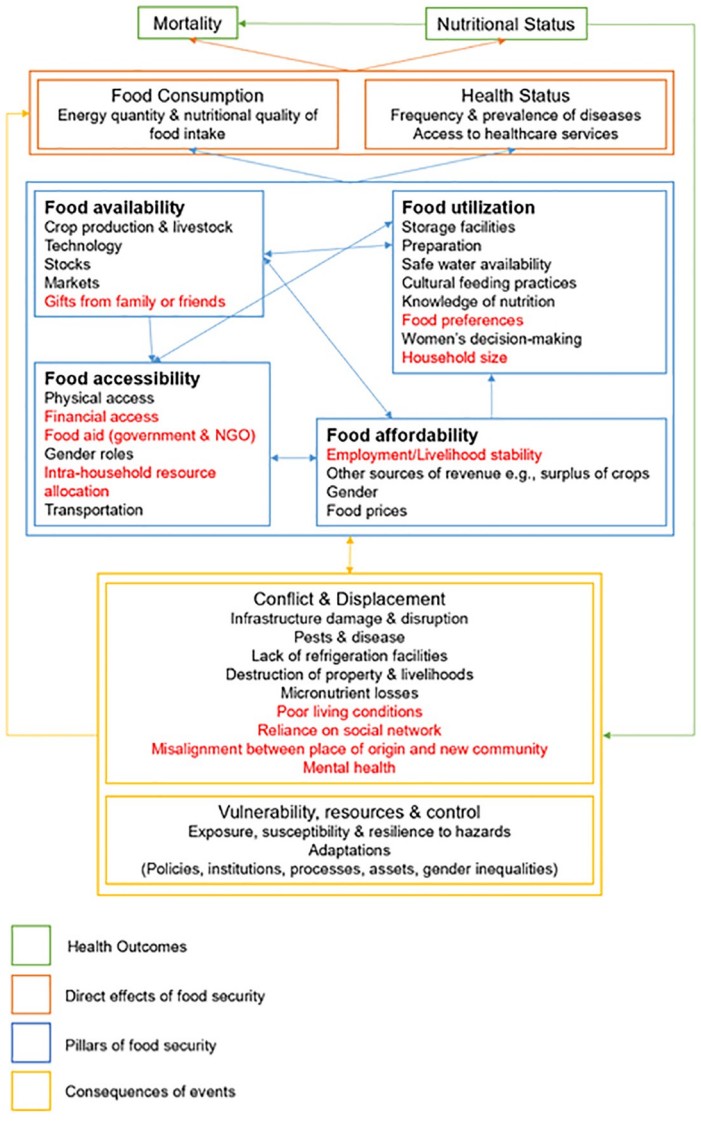

**Fig 5. A conceptual framework analysing the determinants contributing to food security.** This framework details the pillars and how they intersect, as well as how they are affected by the consequences of conflict and displacement. This then has an impact on people's consumption patterns and nutritional status, thus resulting in a feedback loop between health outcomes and vulnerability to future difficulties. Those variables in red have been identified as important for this population.

Most participants (80%) reported some level of unhappiness around the size of their accommodation. A qualitative study conducted in Southern Australia among refugees and asylum seekers exploring the relationship between housing and health revealed that a significant number of participants were worried about the size of the accommodation, which was reported to have a negative impact on their mental health [43]. Many factors affect the wellbeing of refugee groups, and this provides an opportunity for stakeholders to conduct site-specific needs assessments to determine optimum and realistic living conditions alongside the level of satisfaction among refugees.

Refugees are entitled to receive welfare support from local authorities, although this is often difficult due to individuals' lack of familiarity with bureaucratic and regulatory barriers [44]. From our study, only 24.1% of refugees received welfare support with the major source of accessible support coming from Government of Ghana and the United Nations agency, such as UNICEF and UNHCR. This finding indicates the key role played by the host-country in supporting the mandated UN systems to take care of displaced persons during emergencies.

Around half of the participants indicated that they felt safe and welcomed by the host communities. Refugees' sense of safety and security in the host community is a complex issue that varies across different contexts. Studies have shown that host communities' attitudes toward refugees can have a significant impact on their sense of safety and security [45]. Acceptance and assistance from the host community are crucial in ensuring support for refugee populations [46]. However, the perception of safety among refugees is subjective and individual, with many refugees not considering themselves to have found complete safety and protection in their host countries [47].

Our analyses indicate that refugees who migrated from rural areas had higher levels of food security compared with those from urban areas. The majority of participants indicated they were originally from rural areas. Thus, it is possible that rural-dwellers can more easily assimilate themselves with community members for support compared to those from cities who may have a problem adjusting to rural life. This is consistent with an earlier study conducted in Australia, which revealed that refugees from rural communities found it easy assimilating with rural host communities compared to urban refugees [48]. Accessing and utilizing food would look different between urban and rural areas; rural-dwellers will be more accustomed to this, whereas previously urban residents' routines, preferences, and accustom to convenience may be misaligned with their new environment. Some may need more assistance to adjust to the community. There can also be a negative impact upon food security within the host communities themselves, as an influx of refugees can lead to uneven distribution of limited food supplies and stretched availability [45]. Longer term interventions to increase the availability of food within these rural communities would be of benefit, such as increasing existing crop production or providing refugees with the resources to contribute.

In this study, refugees who are presently residing with an acquaintance within the host community, such as a relative or a friend, encountered fewer complexities concerning food insecurity in comparison to individuals residing in housing exclusively designated for refugees or internally displaced persons. These findings are in line with an earlier study reporting the importance of social support and community networking in improving the well-being of refugees [49, 50]. The result equally confirms the trade and inter-personal relationships that exists between the northern part of Ghana and southern Burkina Faso [51]. These relationships are established by historical, cultural, and economic elements, with migration and familial connections present between the populations of both nations [52].

The displacement of Burkina Faso populations are on a much smaller scale than the situation with Ukraine [44], and perhaps this is one of the reasons for the lower profile of this crisis in West Africa. However, the needs and conditions of the populations described here appear to be worse than those experienced by Ukraine refugees who are typically wealthier and moving to higher-income settings. When comparing Burkina Faso refugees in Ghana with the population remaining within Afghanistan, the prevalence of food insecurity is similarly high [53]. Malnutrition is one of the key concerns in Afghanistan, and this has to be a real concern for those addressing the refugee crisis in northern Ghana.

### Strengths and limitations

The strength of this study lies in the fact that this was a community-based cross-sectional study, with high-level refugees' participation. This study fills a knowledge gap around assessment of food insecurity, safety, and wellbeing among refugees within the sub-region.

However, the study contains limitations. The tool that was used to measure household food insecurity provides a direct measure of household ability to afford food within the last 30 days and does not take into account other factors, including seasonality of food production and availability, discrimination in food allocation, and food preference. Whilst this tool is validated for use in lower-income settings, it may lack flexibility to take into account nuances from this specific population. These aspects must be explored further to understand how they influence household food insecurity. There is also little data available around the demographics of this refugee population, and thus we cannot be sure how representative our study populations are. This is also a one-off survey at one point in time–the refugee situation evolves quickly, and repeated similar measurements would provide an up to date assessment of health and social needs, along with an indications around any seasonal impacts. Additional quantitative and qualitative research should highlight how the other environmental factors, including employment opportunities, family size, housing types, and heath conditions, may contribute to our understanding of specific drivers of food insecurity within this vulnerable population.

### Conclusions

The current study found a high prevalence of household food insecurity among the Burkina Faso refugee population residing in Ghana, with refugees staying within host communities reporting higher food security. Additionally, participants dissatisfied with their accommodation size were found to be at higher risk of food insecurity. Despite these challenges, the majority of participants felt safe and welcomed by the host communities. To address these issues, we recommend that governments, UNHCR, and UNICEF improve the logistics of food distribution by increasing the frequency and reliability of deliveries. Exploring partnerships with local suppliers or community-based organizations could help streamline distribution, ensuring refugees receive timely and adequate food supplies. In terms of accommodation, measures should be taken to improve living conditions by providing sufficient shelter sizes and expanding the availability of suitable housing. Sustainable housing solutions, such as modular or prefabricated units, should be considered to meet the growing needs. It is also crucial to consider the needs of surrounding host communities to ensure equitable access to resources, which can help reduce potential tensions or hostility between refugees and local populations.

Healthcare access should be strengthened, particularly for malnutrition and food insecurity-related health issues. Mobile health units could improve accessibility for refugees in remote areas, while local health centers should be supported to handle the increased demand. Additionally, a proactive review of the emerging health burden due to food insecurity should be incorporated into health planning. Finally, we recommend repeating this survey in a longitudinal study to track changes over time and provide up-to-date findings for decision-makers. Conducting similar studies in other refugee settings can help identify similarities and differences across populations, offering a broader understanding of the factors affecting food insecurity, safety perceptions, and social integration.

### Supporting information

**S1 File. Survey administered to the Burkina Faso refugee participants.**
(DOCX)

**S2 File. R markdown document showing the codes required to analyse food insecurity data.**
(PDF)

**S3 File. Confirmation of ethics approval.**
(PDF)

## Acknowledgments

The study team would like to acknowledge and thank the participants and data collectors for their help and support with this research.

## Author Contributions

**Conceptualization:** Abdul-Wahab Inusah, Tahiru Issahaku Ahmed, David Tetteh Nartey, Michael Head, Shamsu-Deen Ziblim.

**Data curation:** David Tetteh Nartey.

**Formal analysis:** Abdul-Wahab Inusah, Ken Brackstone, Jessica L. Boxall, Ashley I. Heinson.

**Funding acquisition:** Michael Head.

**Methodology:** Abdul-Wahab Inusah, Ken Brackstone, David Tetteh Nartey, Michael Head, Shamsu-Deen Ziblim.

**Project administration:** Abdul-Wahab Inusah, Tahiru Issahaku Ahmed, David Tetteh Nartey, Michael Head.

**Software:** Jessica L. Boxall, Ashley I. Heinson.

**Supervision:** Abdul-Wahab Inusah, Tahiru Issahaku Ahmed, David Tetteh Nartey, Shamsu-Deen Ziblim.

**Validation:** Abdul-Wahab Inusah.

**Writing – original draft:** Abdul-Wahab Inusah, Ken Brackstone, Tahiru Issahaku Ahmed, David Tetteh Nartey, Jessica L. Boxall, Ashley I. Heinson, Michael Head, Shamsu-Deen Ziblim.

**Writing – review & editing:** Abdul-Wahab Inusah, Ken Brackstone, Tahiru Issahaku Ahmed, David Tetteh Nartey, Jessica L. Boxall, Ashley I. Heinson, Michael Head, Shamsu-Deen Ziblim.

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
