## [Decision Letter · Decision Letter 0]

14 Nov 2024

PONE-D-24-23199Household food insecurity, living conditions, and individual sense of security: a cross-sectional survey among Burkina Faso refugees in GhanaPLOS ONE

Dear Dr. Head,

Thank you for submitting your manuscript to PLOS ONE. After careful consideration, we feel that it has merit but does not fully meet PLOS ONE’s publication criteria as it currently stands. Therefore, we invite you to submit a revised version of the manuscript that addresses the points raised during the review process.

We look forward to receiving your revised manuscript.

Kind regards,

António Raposo

Academic Editor

PLOS ONE

Journal Requirements:

2. Please note that your Data Availability Statement is currently missing the repository name. If your manuscript is accepted for publication, you will be asked to provide these details on a very short timeline. We therefore suggest that you provide this information now, though we will not hold up the peer review process if you are unable.

3. We note that Figure 1 in your submission contain map/satellite images which may be copyrighted. All PLOS content is published under the Creative Commons Attribution License (CC BY 4.0), which means that the manuscript, images, and Supporting Information files will be freely available online, and any third party is permitted to access, download, copy, distribute, and use these materials in any way, even commercially, with proper attribution. For these reasons, we cannot publish previously copyrighted maps or satellite images created using proprietary data, such as Google software (Google Maps, Street View, and Earth). For more information, see our copyright guidelines: http://journals.plos.org/plosone/s/licenses-and-copyright. We require you to either (1) present written permission from the copyright holder to publish these figures specifically under the CC BY 4.0 license, or (2) remove the figures from your submission:

a. You may seek permission from the original copyright holder of Figure 1 to publish the content specifically under the CC BY 4.0 license. We recommend that you contact the original copyright holder with the Content Permission Form (http://journals.plos.org/plosone/s/file?id=7c09/content-permission-form.pdf) and the following text: “I request permission for the open-access journal PLOS ONE to publish XXX under the Creative Commons Attribution License (CCAL) CC BY 4.0 (http://creativecommons.org/licenses/by/4.0/). Please be aware that this license allows unrestricted use and distribution, even commercially, by third parties. Please reply and provide explicit written permission to publish XXX under a CC BY license and complete the attached form.” Please upload the completed Content Permission Form or other proof of granted permissions as an "Other" file with your submission. In the figure caption of the copyrighted figure, please include the following text: “Reprinted from [ref] under a CC BY license, with permission from [name of publisher], original copyright [original copyright year].”

b. If you are unable to obtain permission from the original copyright holder to publish these figures under the CC BY 4.0 license or if the copyright holder’s requirements are incompatible with the CC BY 4.0 license, please either i) remove the figure or ii) supply a replacement figure that complies with the CC BY 4.0 license. Please check copyright information on all replacement figures and update the figure caption with source information. If applicable, please specify in the figure caption text when a figure is similar but not identical to the original image and is therefore for illustrative purposes only. The following resources for replacing copyrighted map figures may be helpful: USGS National Map Viewer (public domain): http://viewer.nationalmap.gov/viewer/ The Gateway to Astronaut Photography of Earth (public domain): http://eol.jsc.nasa.gov/sseop/clickmap/ Maps at the CIA (public domain): https://www.cia.gov/library/publications/the-world-factbook/index.html and https://www.cia.gov/library/publications/cia-maps-publications/index.html NASA Earth Observatory (public domain): http://earthobservatory.nasa.gov/ Landsat: http://landsat.visibleearth.nasa.gov/ USGS EROS (Earth Resources Observatory and Science (EROS) Center) (public domain): http://eros.usgs.gov/# Natural Earth (public domain): http://www.naturalearthdata.com/  

Reviewers' comments:

Reviewer's Responses to Questions

**Comments to the Author**

1. Is the manuscript technically sound, and do the data support the conclusions?

Reviewer #1: Partly

Reviewer #2: Yes

Reviewer #3: Yes

Reviewer #4: Yes

2. Has the statistical analysis been performed appropriately and rigorously? 

Reviewer #1: Yes

Reviewer #2: Yes

Reviewer #3: Yes

Reviewer #4: Yes

3. Have the authors made all data underlying the findings in their manuscript fully available?

Reviewer #1: Yes

Reviewer #2: Yes

Reviewer #3: Yes

Reviewer #4: Yes

4. Is the manuscript presented in an intelligible fashion and written in standard English?

Reviewer #1: Yes

Reviewer #2: Yes

Reviewer #3: Yes

Reviewer #4: Yes

5. Review Comments to the Author

Reviewer #1: This study provides interesting insights into the significant food insecurity challenges facing Burkina Faso refugees in Ghana’s Upper East region.

As a one-off cross-sectional survey, the study captures food insecurity at a single point in time, which may not accurately reflect fluctuations in food access or conditions over time. A longitudinal approach would be more appropriate for tracking how food insecurity levels change, particularly given the volatile nature of refugee populations and their reliance on aid.

Although 498 households were surveyed, the study does not specify how representative this sample is of the overall Burkina Faso refugee population in northern Ghana. More information on sampling techniques would help clarify whether these results can be generalized across different refugee populations or other regions.

The study relied on community-focal persons to contact registered refugee households, which may introduce selection bias. Refugees more integrated into the community or more familiar with focal persons may have been overrepresented, potentially skewing the results on food insecurity, social integration, and perceptions of safety.

While the Rasch model is a reliable method for scaling responses, it would be helpful to see more justification for its use in this context. Additionally, further explanation of how Rasch scores were interpreted in assessing food insecurity levels would improve clarity. Complementary methods, such as factor analysis or other model checks, could strengthen the robustness of findings.

While the study highlights key factors (e.g., urban vs. rural origin, camp vs. community living), it lacks consideration of potentially confounding variables, such as employment opportunities, family size, and health conditions. Controlling for these variables would provide a clearer understanding of the specific drivers of food insecurity within this population.

The link between accommodation dissatisfaction and food insecurity was marginally significant (p = .060), suggesting only a weak association. This should be interpreted cautiously. Additional qualitative insights, such as interviews, could shed light on how living conditions contribute to the experience of food insecurity and whether dissatisfaction stems from factors like overcrowding, location, or infrastructure inadequacies.

Although the study calls for improved food distribution logistics, accommodation facilities, and healthcare access, these recommendations lack specificity. For instance, outlining what logistical improvements are necessary (e.g., increased food delivery frequency, local partnerships) or how accommodation can be sustainably expanded would enhance the applicability of these findings for stakeholders like governments, UNHCR, and UNICEF.

Finding that 73.5% of refugees feel safe and welcomed is encouraging, but it would benefit from further investigation into what factors influence these perceptions, particularly as food insecurity often increases vulnerability. A more nuanced analysis might also explore how safety perceptions differ between camps and community-integrated refugees.

Reviewer #2: I commend the authors on a capably conducted and important study of food insecurity in a refugee context. I recommend the following revisions.

1. The paper would be enhanced with some theoretical context and consideration of possible theoretical implications. I recommend food and nutrition security theory, which can be accessed at the following link: https://journals.sagepub.com/doi/10.1177/0379572120925341#:~:text=Food%20and%20nutrition%20insecurity%20continue,but%20not%20acceptable%20or%20sustainable. This theory can provide greater analytical precision. Perhaps most importantly, the authors’ empirical investigation could be used to advance and refine the theory given the refugee context they are studying. That could be done in the Discussion.

2. I appreciate the authors’ attention to methodological detail. Were power analyses needed or employed to arrive at the sufficient sample size number identified in the manuscript? They have addressed sample size considerations using a single population proportion formula. I’d welcome another couple of sentences on this front for additional clarity.

3. There were two stages of sampling used here. First, purposive sampling was used for community selection, which is completely understandable. Second, individuals were sampled prior to survey administration. Should we be concerned about selectivity bias in this second stage of sampling? Could reliance on the community refugee focal person lead to sampling bias due to the manner in which this work is done? More notably, could some refugees decline or opt out in a nonrandom fashion? Please address these possibilities if they are reasonable threats. They do not undermine the study but could be limitations.

4. I like that religion (faith tradition) was examined as a control. Please expound briefly on traditional religion, though I’m pretty sure I understand that. Also, was level of religious activity gauged? If so, I’d say to test for that.

5. Some proofing for spacing would be welcome. See, e.g., the first sentence of the second Discussion paragraph. I saw this in several spaces.

Generally, this is an impressive study with important findings. Well done.

Reviewer #3: Comments:

The manuscript addresses a relevant research question within the journal's scope, examining household food insecurity, living conditions, and perceptions of security among Burkina Faso refugees residing in the Upper East region of Ghana.

To enhance the manuscript's structure and flow, I have provided some comments, questions, and suggestions below.

The title effectively captures the manuscript's focus and is concise. The abstract accurately summarizes the objectives, methods, results, and conclusions, though some revisions are needed for clarity.

Throughout the text:

- I recommend using justified alignment for uniform spacing.

- Maintain a consistent font throughout the text, including in the references.

- Standardize references to supplementary materials to avoid variations, for instance, using either “(supplementary S1),” “supplementary information (S2),” or “(Supplementary, S3).”

Abstract:

- Replace “key words” with “keywords.”

Introduction:

- I suggest adding a paragraph after “Refugees often flee from armed conflict [...].”

- Provide the full name of UNICEF at its first mention. Abbreviations should be introduced only upon their first use.

Methods:

- Sampling technique and procedures: Include a separate paragraph describing inclusion and exclusion criteria, beginning with “Inclusion criteria were adults [...].”

- Study variables: If describing subsections, avoid using bold on the same line as text; consider adding a paragraph before each subsection.

- Sense of security: In “1 = strongly disagree; 5 = strongly agree,” consider removing italics.

- Data collection procedure and quality control: Add a paragraph after “Ten community health workers participated [...].”

- Briefly outline FIES 1-8 variables in the Data analysis techniques section.

Results:

- Tables 1 and 3: Replace “yrs” with “years.”

- Provide the full name of FAO upon its first mention, as abbreviations should be introduced only once.

- Tables 2 and 3: Match the format of these tables with Table 1 regarding borders and lines.

- Table 5: Insert a line before explaining the asterisk referring to the average to clarify that it belongs to the legend.

- Ensure consistency in table legends, using either italics or plain text.

Discussion:

- Add references to support or contrast with the findings in the paragraph discussing Figure 5: “The conceptual framework in Figure 5 is adapted from adapted [...].”

- Consider adding references in the paragraph, “Around half of the participants in this study resided in refugee camps set up by UNHCR and the local authorities [...].”

- Add references to support the paragraph: “Refugees are entitled to receive welfare support from local authorities [...].”

- Add a paragraph after “In this study, refugees presently residing with an acquaintance within the host community [...].”

- Add references for the paragraph: “The displacement of Burkina Faso populations is on a much smaller scale than the situation with Ukraine [...].”

Strengths and Limitations:

- Add a paragraph after “However, the study contains limitations [...].”

Conclusion:

- Add a paragraph following “The short-term measures could include improved logistics [...].”

References:

- Use brackets instead of parentheses for in-text citations.

- Consistently place references in brackets after the period at the end of the sentence; some currently appear before the period.

- Ensure the font and size of references match the rest of the text.

- Format references in Vancouver style, per the author guidelines.

Figures:

- Improve the quality of images.

- Figure 4: Explain the abbreviation CSOs in the caption.

Supplementary Material:

- S1: Format the supplementary material to use the same font, color, and size as the main text, with bold and italics consistent across all items. Some items currently appear in different fonts, colors, sizes, and styles.

Reviewer #4: The study is highly relevant and was conducted in a robust and appropriate manner. The results obtained by the authors represent a significant contribution. However, the discussion falls short of the quality presented in the rest of the manuscript. After reading the results, it is natural for readers to expect a discussion on the determinants that contributed to the food security observed among Burkinabe refugees in Ghana. Yet, the authors address these determinants only briefly. Numerous comparisons are made with refugees in various other countries, which have socioeconomic conditions and refugee support systems quite different from those in Ghana. By relying on multiple isolated citations, the authors miss an opportunity to focus on the key determinants of food insecurity, which would allow readers a deeper understanding of the complex situation faced by refugees in Ghana.

6. PLOS authors have the option to publish the peer review history of their article (what does this mean?). If published, this will include your full peer review and any attached files.

Reviewer #1: **Yes: **M. João Reis Lima

Reviewer #2: No

Reviewer #3: **Yes: **Marcela Gomes Reis

Reviewer #4: No

---

## [Author Response · Author response to Decision Letter 0]

16 Dec 2024

If easier, see the file attached to the submission. The text is also pasted in entirety here too.

Author response to reviewer and editorial comments

14 December 2024

Thank you to reviewers and colleagues at PLOS for the opportunity to further revise this manuscript. 

See below, in blue italics, for the reviewer/editor comment, point by point. 

And plain text for our author response.

Editorial feedback

We have made some revisions accordingly, e.g changing key headings to level 21 format (size 18 font), revising names of figure files to simply ‘Fig 1’ etc. 

If any further formatting changes are required, we would be pleased to make them.

2. Please note that your Data Availability Statement is currently missing the repository name. If your manuscript is accepted for publication, you will be asked to provide these details on a very short timeline. We therefore suggest that you provide this information now, though we will not hold up the peer review process if you are unable.

We have revised the data availability statement in the submission portal to the following - 

“The dataset is publicly available on the Figshare repository at https://doi.org/10.6084/m9.figshare.23782893”

3. We note that Figure 1 in your submission contain map/satellite images which may be copyrighted. All PLOS content is published under the Creative Commons Attribution License (CC BY 4.0), which means that the manuscript, images, and Supporting Information files will be freely available online, and any third party is permitted to access, download, copy, distribute, and use these materials in any way, even commercially, with proper attribution. For these reasons, we cannot publish previously copyrighted maps or satellite images created using proprietary data, such as Google software (Google Maps, Street View, and Earth). For more information, see our copyright guidelines: http://journals.plos.org/plosone/s/licenses-and-copyright. We require you to either (1) present written permission from the copyright holder to publish these figures specifically under the CC BY 4.0 license, or (2) remove the figures from your submission:

a. You may seek permission from the original copyright holder of Figure 1 to publish the content specifically under the CC BY 4.0 license. We recommend that you contact the original copyright holder with the Content Permission Form (http://journals.plos.org/plosone/s/file?id=7c09/content-permission-form.pdf) and the following text: “I request permission for the open-access journal PLOS ONE to publish XXX under the Creative Commons Attribution License (CCAL) CC BY 4.0 (http://creativecommons.org/licenses/by/4.0/). Please be aware that this license allows unrestricted use and distribution, even commercially, by third parties. Please reply and provide explicit written permission to publish XXX under a CC BY license and complete the attached form.” Please upload the completed Content Permission Form or other proof of granted permissions as an "Other" file with your submission. In the figure caption of the copyrighted figure, please include the following text: “Reprinted from [ref] under a CC BY license, with permission from [name of publisher], original copyright [original copyright year].”

b. If you are unable to obtain permission from the original copyright holder to publish these figures under the CC BY 4.0 license or if the copyright holder’s requirements are incompatible with the CC BY 4.0 license, please either i) remove the figure or ii) supply a replacement figure that complies with the CC BY 4.0 license. Please check copyright information on all replacement figures and update the figure caption with source information. If applicable, please specify in the figure caption text when a figure is similar but not identical to the original image and is therefore for illustrative purposes only. The following resources for replacing copyrighted map figures may be helpful: USGS National Map Viewer (public domain): http://viewer.nationalmap.gov/viewer/ The Gateway to Astronaut Photography of Earth (public domain): http://eol.jsc.nasa.gov/sseop/clickmap/ Maps at the CIA (public domain): https://www.cia.gov/library/publications/the-world-factbook/index.html and https://www.cia.gov/library/publications/cia-maps-publications/index.html NASA Earth Observatory (public domain): http://earthobservatory.nasa.gov/ Landsat: http://landsat.visibleearth.nasa.gov/ USGS EROS (Earth Resources Observatory and Science (EROS) Center) (public domain): http://eros.usgs.gov/# Natural Earth (public domain): http://www.naturalearthdata.com/

We want to clarify that the map/satellite image is entirely the Authors’ creation. The data used to generate the figure was part of the primary data collected during the study’s data collection phase. The figure was created using QGIS, an open-source Geographic Information System (GIS) tool, and does not incorporate any proprietary or copyrighted images, such as those from Google Maps, Google Earth, or similar sources.

Since the figure is entirely based on our original work and generated using open-source software, we kindly request that it be considered for publication under the Creative Commons Attribution License (CC BY 4.0)

Reviewer 1

Reviewer #1: This study provides interesting insights into the significant food insecurity challenges facing Burkina Faso refugees in Ghana’s Upper East region.

As a one-off cross-sectional survey, the study captures food insecurity at a single point in time, which may not accurately reflect fluctuations in food access or conditions over time. A longitudinal approach would be more appropriate for tracking how food insecurity levels change, particularly given the volatile nature of refugee populations and their reliance on aid.

We do already highlight the cross-sectional nature of this study in the limitations paragraph. However, we have revised the conclusions section to include this sentence - 

“Further research can include repeating this survey within a longitudinal study to ensure up-to-date findings and temporal trends are available for decision-makers”

Although 498 households were surveyed, the study does not specify how representative this sample is of the overall Burkina Faso refugee population in northern Ghana. More information on sampling techniques would help clarify whether these results can be generalized across different refugee populations or other regions.

Thank you for your comments: The total number of refugees within the study area was 1679. The number of registered refugees for each district was 1072 in Binduri District (64%), 495 in Bawku Municipal (29%) and 112 in Bawku West district (7%). The sample size was calculated using a single population proportion formula with an assumption of 50% population proportion, with a 95% confidence level, a 5% margin of error, and a normal population distribution Z = 1.96, we adjusted for a 5% non-response rate to arrive at 400. 

For more detail, see the sections “Study population and sample size determination” and “Sampling technique and sampling procedures”.

The study relied on community-focal persons to contact registered refugee households, which may introduce selection bias. Refugees more integrated into the community or more familiar with focal persons may have been overrepresented, potentially skewing the results on food insecurity, social integration, and perceptions of safety.

Thank you for your thoughtful comments regarding the potential for selection bias in our study. We appreciate your insights and agree that this is an important consideration.

However, we would like to clarify the context of our study. The research was conducted a few weeks after the refugees’ arrival, during which time they were officially registered by community-focal persons upon arrival. This registration process occurred before refugees were allocated shelters, either in tents or with community members. As such, the initial registration included all refugees arriving at that time, providing a comprehensive and representative sampling frame.

To mitigate any potential overrepresentation of certain groups, such as those more integrated into the community or more familiar with the focal persons, we relied on the official list of registered refugees to guide our sampling. This supports our approaches with capturing a balanced representation of the refugee population, and minimizes the risk of selection bias in the reported findings on food insecurity, social integration, and perceptions of safety.

While the Rasch model is a reliable method for scaling responses, it would be helpful to see more justification for its use in this context. Additionally, further explanation of how Rasch scores were interpreted in assessing food insecurity levels would improve clarity. Complementary methods, such as factor analysis or other model checks, could strengthen the robustness of findings.

We have revised text in the methods, sub-section data analysis techniques, it now reads as follows - “Rasch modelling was undertaken in R 4.3.2 using the package “RM.Weights”, as the recommended analysis protocol by the FAO.[21]”

While the study highlights key factors (e.g., urban vs. rural origin, camp vs. community living), it lacks consideration of potentially confounding variables, such as employment opportunities, family size, and health conditions. Controlling for these variables would provide a clearer understanding of the specific drivers of food insecurity within this population.

Many thanks for your comment. We agree that including additional variables such as employment, family size, and health conditions, could help to provide a clearer understanding of the specific drivers of food insecurity. 

Unfortunately, we were unable to delve into these questions with the survey that we had. We designed the survey to be shorter so as not to overwhelm participants with too many questions. However, this would be great to conduct in the future to flesh out the nuances of these relationships. We have therefore referenced this in the discussion section, and commented about how future research could control for these potentially confounding variables.

The link between accommodation dissatisfaction and food insecurity was marginally significant (p = .060), suggesting only a weak association. This should be interpreted cautiously. Additional qualitative insights, such as interviews, could shed light on how living conditions contribute to the experience of food insecurity and whether dissatisfaction stems from factors like overcrowding, location, or infrastructure inadequacies.

Agreed that a qualitative study would be useful to consider more fully the ‘lived experience’ of these populations, and to build upon our quantitative work here.

Thus, we have added this into the limitations section - 

“Additional qualitative research could highlight how the environment, including living conditions, may contribute to the food insecurity and perceived contributions from factors such as overcrowding, type of housing, or family size.”

Although the study calls for improved food distribution logistics, accommodation facilities, and healthcare access, these recommendations lack specificity. For instance, outlining what logistical improvements are necessary (e.g., increased food delivery frequency, local partnerships) or how accommodation can be sustainably expanded would enhance the applicability of these findings for stakeholders like governments, UNHCR, and UNICEF.

We appreciate your suggestion to include more specificity in the recommendations, and we have revised them accordingly Below is the revised conclusions section

“The current study found a high prevalence of household food insecurity among the Burkina Faso refugee population residing in Ghana, with refugees staying within host communities reporting higher food security. Additionally, participants dissatisfied with their accommodation size were found to be at higher risk of food insecurity. Despite these challenges, the majority of participants felt safe and welcomed by the host communities.

To address these issues, we recommend that governments, UNHCR, and UNICEF improve the logistics of food distribution by increasing the frequency and reliability of deliveries. Exploring partnerships with local suppliers or community-based organizations could help streamline distribution, ensuring refugees receive timely and adequate food supplies.

In terms of accommodation, measures should be taken to improve living conditions by providing sufficient shelter sizes and expanding the availability of suitable housing. Sustainable housing solutions, such as modular or prefabricated units, should be considered to meet the growing needs. It is also crucial to consider the needs of surrounding host communities to ensure equitable access to resources, which can help reduce potential tensions or hostility between refugees and local populations.

Healthcare access should be strengthened, particularly for malnutrition and food insecurity-related health issues. Mobile health units could improve accessibility for refugees in remote areas, while local health centers should be supported to handle the increased demand. Additionally, a proactive review of the emerging health burden due to food insecurity should be incorporated into health planning.

Finally, we recommend repeating this survey in a longitudinal study to track changes over time and provide up-to-date findings for decision-makers. Conducting similar studies in other refugee settings can help identify similarities and differences across populations, offering a broader understanding of the factors affecting food insecurity, safety perceptions, and social integration.”

We trust these more detailed recommendations will better guide future interventions and ensure that the findings are actionable for key stakeholders.

Finding that 73.5% of refugees feel safe and welcomed is encouraging, but it would benefit from further investigation into what factors influence these perceptions, particularly as food insecurity often increases vulnerability. A more nuanced analysis might also explore how safety perceptions differ between camps and community-integrated refugees.

Thank you for your insightful feedback. We agree that exploring the factors influencing refugees' perceptions of safety and how these perceptions differ between camps and community-integrated settings could provide valuable insights. 

At present, our data set does not include a detailed comparison between camps and community settings regarding safety perceptions. However, we recognize the importance of this distinction and its potential implications. We plan to include this as a focal point in future research, as well as examine how intersecting vulnerabilities like food insecurity might influence safety perceptions.

We appreciate your suggestion, as it highlights opportunities to expand on our findings and enrich the study's implications.

Reviewer 2

Reviewer #2: I commend the authors on a capably conducted and important study of food insecurity in a refugee context. I recommend the following revisions.

1. The paper would be enhanced with some theoretical context and consideration of possible theoretical implications. I recommend food and nutrition security theory, which can be accessed at the following link: https://journals.sagepub.com/doi/10.1177/0379572120925341#:~:text=Food%20and%20nutrition%20insecurity%20continue,but%20not%20acceptable%20or%20sustainable. This theory can provide greater analytical precision. Perhaps most importantly, the authors’ empirical investigation could be used to advance and refine the theory given the refugee context they are studying. That could be done in the Discussion.

We briefly discuss the theoretical framework in in the introdu

---

## [Decision Letter · Decision Letter 1]

29 Dec 2024

Household food insecurity, living conditions, and individual sense of security: a cross-sectional survey among Burkina Faso refugees in Ghana

PONE-D-24-23199R1

Dear Dr. Head,

We’re pleased to inform you that your manuscript has been judged scientifically suitable for publication and will be formally accepted for publication once it meets all outstanding technical requirements.

Kind regards,

António Raposo

Academic Editor

PLOS ONE

Additional Editor Comments (optional):

Reviewers' comments:

Reviewer's Responses to Questions

**Comments to the Author**

1. If the authors have adequately addressed your comments raised in a previous round of review and you feel that this manuscript is now acceptable for publication, you may indicate that here to bypass the “Comments to the Author” section, enter your conflict of interest statement in the “Confidential to Editor” section, and submit your "Accept" recommendation.

Reviewer #1: All comments have been addressed

Reviewer #2: All comments have been addressed

2. Is the manuscript technically sound, and do the data support the conclusions?

Reviewer #1: Yes

Reviewer #2: Yes

3. Has the statistical analysis been performed appropriately and rigorously? 

Reviewer #1: (No Response)

Reviewer #2: Yes

4. Have the authors made all data underlying the findings in their manuscript fully available?

Reviewer #1: Yes

Reviewer #2: Yes

5. Is the manuscript presented in an intelligible fashion and written in standard English?

Reviewer #1: Yes

Reviewer #2: Yes

6. Review Comments to the Author

Reviewer #1: Considering the answers and the modifications made in the manuscript by the authors, I believe it can now be published.

Reviewer #2: I commend the authors on a sound and thorough revision. In my view, this paper is ready to move forward.

7. PLOS authors have the option to publish the peer review history of their article (what does this mean?). If published, this will include your full peer review and any attached files.

Reviewer #1: **Yes: **M. João Reis Lima

Reviewer #2: No

---

## [Editor Report · Acceptance letter]

2 Jan 2025

PONE-D-24-23199R1 

PLOS ONE

Dear Dr. Head, 

I'm pleased to inform you that your manuscript has been deemed suitable for publication in PLOS ONE. Congratulations! Your manuscript is now being handed over to our production team.

Kind regards, 

on behalf of

Dr. António Raposo 

Academic Editor

PLOS ONE